# Social media addiction and emotions during the disaster recovery period—The moderating role of post-COVID timing

Dewan Muhammad Nur –A Yazdani[1]*, Tanvir Abir[2], Yang Qing[3], Jamee Ahmad[4], Abdullah Al Mamun[5], Noor Raihani Zainol[6], Kaniz Kakon[7], Kingsley Emwinyore Agho[8,9], Shasha Wang[10]

1 MONASH Pathway at Universal College Bangladesh, Dhaka, Bangladesh, 2 Department of Business Administration, Faculty of Business & Entrepreneurship (FBE), Daffodil International University, Dhaka, Bangladesh, 3 UCSI Graduate Business School, UCSI University, Kuala Lumpur, Malaysia, 4 College of Business Administration—CBA, International University of Business, Agriculture and Technology—IUBAT, Dhaka, Bangladesh, 5 UKM-Graduate School of Business, Universiti Kebangsaan Malaysia, Bangi, Selangor Darul Ehsan, Malaysia, 6 Faculty of Entrepreneurship and Business, Universiti Malaysia Kelantan, Kota Bharu, Malaysia, 7 Department of Philosophy, College of Arts and Sciences—CAAS, International University of Business Agriculture and Technology—IUBAT, Dhaka, Bangladesh, 8 School of Health Sciences, Western Sydney University, Campbelltown, New South Wales, Australia, 9 African Vision Research Institute, Discipline of Optometry, University of KwaZulu-Natal, Durban, South Africa, 10 Marketing and Public Relations at the QUT Business School, Queensland University of Technology, Queensland, Australia

* dewan.yazdani@ucbbd.org, dewanyazdani@gmail.com

**Data Availability Statement:** All relevant data are within the manuscript and its Supporting information files.

**Funding:** The authors received no specific funding for this work.

## Abstract

### Background

Social media addiction, a recently emerged term in medical science, has attracted the attention of researchers because of its significant physical and psychological effects on its users. The issue has attracted more attention during the COVID era because negative emotions (e.g., anxiety and fear) generated from the COVID pandemic may have increased social media addiction. Therefore, the present study investigates the role of negative emotions and social media addiction (SMA) on health problems during and after the COVID lockdown.

### Methods

A survey was conducted with 2926 participants aged between 25 and 45 years from all eight divisions of Bangladesh. The data collection period was between 2nd September– 13th October, 2020. Partial Least Square Structural Equation Modelling (PLS-SEM) was conducted for data analysis by controlling the respondents' working time, leisure time, gender, education, and age.

### Results

Our study showed that social media addiction and time spent on social media impact health. Interestingly, while anxiety about COVID increased social media addition, fear about COIVD reduced social media addition. Among all considered factors, long working hours

**Competing interests:** The authors have declared that no competing interests exist.

contributed most to people's health issues, and its impact on social media addiction and hours was much higher than negative emotions. Furthermore, females were less addicted to social media and faced less health challenges than males.

## Conclusion

The impacts of negative emotions generated by the COVID disaster on social media addiction and health issues should be reconsidered. Government and employers control people's working time, and stress should be a priority to solve people's social media addiction-related issues.

## Introduction

Social media, being a fundamental part of people's lives, leaves an immense impact on every aspect. Scrolling and checking social media has become almost a daily routine of over half of the world's population's daily activities. According to statistics, in the last five years, the number of social media users has almost doubled, increasing from 2.2 billion in 2015 to 4.5 billion in 2022 [1]. This number of users is increasing at an incredible rate. In addition, according to data, there will be 52.58 million internet subscribers in Bangladesh in January 2022. At the beginning of 2022, Bangladesh's number of internet users was 31.5% population. According to Kepios, internet subscribers in Bangladesh rose by 5.5 million (+11.6%) between 2021 and 2022. At the beginning of 2022, 29.7% of Bangladesh's overall inhabitants used social media. Furthermore, according to data revealed in Meta's advertising materials, Facebook had 44.70 million subscribers in Bangladesh in January 2022. According to Google's commercial techniques, YouTube had 34.50 million subscribers in Bangladesh as of early 2022. As a result, it is pretty much evident that the usage of internet in Bangladesh is increasing on a regular basis [2]. The number is rising because social media is the only web-based platform where people with similar backgrounds, interests, activities, and connections can be linked [3]. Besides, the financial ability of people to buy social media (Facebook, Snapchat, Twitter, WhatsApp, Instagram etc.) accessible devices such as smartphones and laptops has increased, resulting in the increasing number of social media users [4]. As social media creates ample opportunities for people to correspond virtually, temporal and partial boundaries notwithstanding [5] and promotes communication and sharing of images and videos amongst social network users, individuals of all ages around the world are taking this advantage [6].

Moreover, the ongoing COVID-19 pandemic caused by Novel Corona Virus significantly influences every individual's lifestyle [7, 8]. Various governments are adopting policies such as lockdown, quarantine etc., to curb the spread of the virus by keeping people indoors [9]. Working from home and virtual education practices have forced people to spend a long time on social media in order to fulfil their needs for work- and disaster-related information, entertainment, and interpersonal communication [10]. Despite the fact that social media plays an undeniably beneficial role in sustaining contact and relationships among individuals, its increased usage is sufficient to result in addiction [11]. A past study has revealed that social media addiction depends on the daily time devoted to the social media platform, and more frequent daily visits increase addiction to social media profiles [12]. Another study found an important association between high school students' daily average internet usage time and social media addiction [13]. However, while social media usage may not always be harmful, some people get addicted and use it extremely or obsessively [14]. Experts have observed the harmful effects of long-term addiction and extreme and obsessive social media usage and

showed that such a level of social media addiction might result in psychological, physiological and productivity issues [15].

There is evidence that obsessive social media usage can impact users' psychological, cognitive, perceptual, and physiological wellbeing. Addicted social media users may have withdrawal feelings, relational issues and others as well [16]. Recent research has found that many people have become addicted to social media due to the COVID-19 epidemic [11], creating several physical health issues such as headaches, sleeping disorders, stomach ailments, and exhaustion [17, 18]. It has been found that using different online platforms, especially social media and shopping websites–whether for essential items or shopping items has increased a lot during COVID-19. This change in the excessive use of digital media has brought numerous physical disorders [19] and left adverse effects on the usual physical activities among the general population [20]. It has been established that despite social media being an essential and integral part of people's lives for day-to-day work and communication, the impact caused by its extensive use on health cannot be ignored [21].

Social media addiction may create significant emotional problems as well. Fear resulting from much information on social media regarding the coronavirus disease and the 'lockdown' situation caused high levels of uncertainty. It raised the level of stress, anxiety, and depression (sometimes leading to suicides) among people worldwide [22–24].

Social media usage hours are strongly correlated with creating social media addiction, and social media addiction causes several physical and psychological issues. Past studies from Bangladesh have assessed the effect of social media and smartphone use vis-à- COVID-19 (the virus). For instance, a study by Islam and colleagues [25] investigated complicated smartphone use, and complex social media use among College and University students in Bangladesh during the COVID-19 pandemic. Findings of that study indicated that problematic social media use was linked to poor psychological wellbeing (such as anxiety and depression) and other factors (particularly landowner age, and poor sleep) during the pandemic, which further suggested the demand for interventions included virtual awareness programmes among College and University students. The present study mainly focused on social media users' health and psychological problems among workers across all the nine districts of Bangladesh after the COVID-19 restriction period imposed by the government of Bangladesh. This study will inform future similar studies and the establishment of new policies seeking to find out how Bangladeshi workers might be affected by social media use during the COVID-19 pandemic. Additionally, the study's results could assist efforts to disseminate behavioural health information on social media.

Reviewing the relevant literature established that social media addiction brings physical and psychological changes. Consequently, the present research focused on psychological and physical issues related to social media addiction. Based on the background of the study above, the following research hypotheses were propounded.

## Conceptual framework

According to the background and objectives of the study and based on the research hypothesises, the researchers have identified the variables of the present study and showed their hypothesised relationship in Fig 1.

## Hypotheses

Based on the research purposes, conceptual framework and the discussion above, the current study established the following research hypotheses:

**Hypothesis 1 (H1)**: Anxiety is correlated with social media addiction.

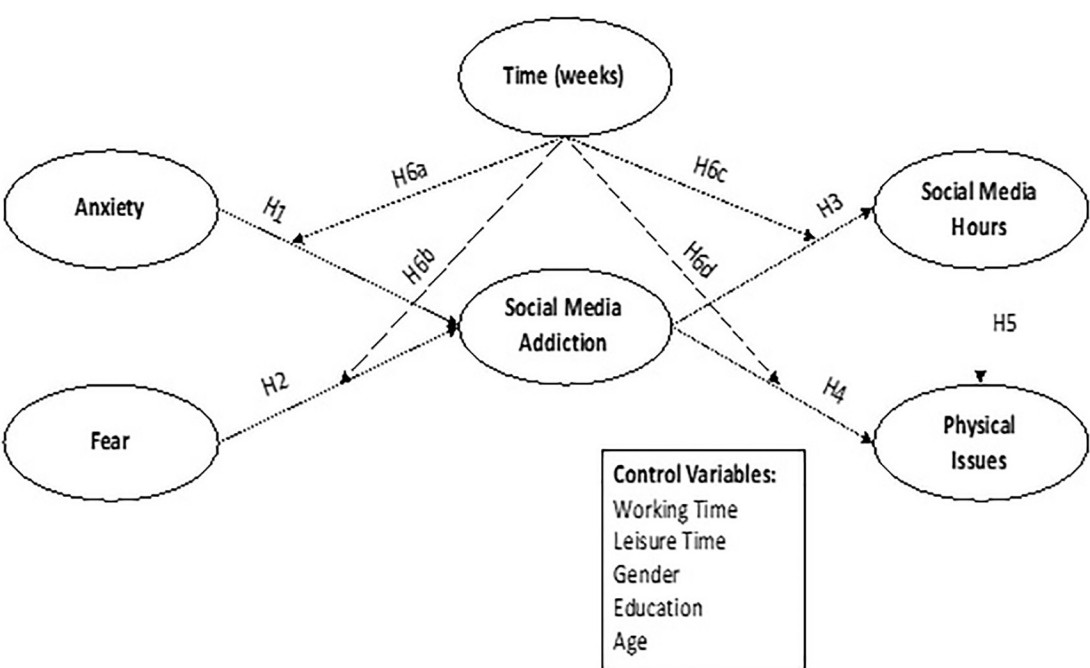

**Fig 1. Conceptual framework for the model (based on Brunborg and Burdzovic, 2019 [26]; Coyne et al., 2020 [27]).**

The COVID-19 pandemic is an epidemiologic and health crisis since it causes extensive psychological issues such as stress, anxiety, depression, trauma, panic, insomnia, death distress, anger, psychosis, boredom, and suicide [28–32].

**Hypothesis 2 (H2)**: Fear is correlated with social media addiction.

However, the pandemic has underscored the downside of social media by showing that uncontrolled use propagates panic, fear, and misinformation about COVID-19 among mass populations [33–35]. There has been a great fear of contracting the virus among many people worldwide bee of the worldwide rise in the death toll due to the virus [36]. Excessive social media use has been identified as one of the major reasons behind this rise in fear. Social media platforms have become home to atrocious and sometimes erroneous information associated with the virus [9]. Social media users spread rumours, conspiracy theories, and even inaccurate calculations of COVID-19 cases and deaths, propagating fear among the masses [33–35].

**Hypothesis 3 (H3)**: Social media addiction is correlated with social media usage hours.

According to a past study [12], social media addiction has risen significantly, as has the amount spent online on a routine basis. The obsession is exacerbated by the more frequent daily visits to social media accounts.

**Hypothesis 4 (H4)**: Social media addiction is correlated with physical health issues.

Past research has revealed that uncontrollable usage of social media affects physical and mental health, such as cardio-metabolic health, sleep, affect, self-esteem, wellbeing and functioning, particularly in adolescents [37].

**Hypothesis 5 (H5)**: Social media usage hours are correlated with physical health issues.

It is evident, and this is inspired by research that 'internet addiction' is principally linked to increased social media or gaming activities [38].

**Hypothesis 6 (H6)**: Time plays a moderating role in the proposed relationships. Specifically, a) the relationship between anxiety and social media addiction will be more substantial over time; b) the relationship between fear and social media addiction will be more substantial over time; c) the relationship between social media addiction and social media usage hours will be more robust over time and d) the relationship between social media addiction and physical issues will be more potent over time, after the disaster.

## Method

### Participants

This study embraced a sequential cross-sectional design. Data were collected for six weeks from 2nd September– 13Th of October, 2020. This period was chosen because of the sad change in people's lifestyle of being isolated from the outside world due to the pandemic and obtaining most of their necessities using a virtual medium. Furthermore, during this period, the situation was getting normal, participants resumed their work physically, and offices opened after the COVID-19 lockdown. Consequently, a new normal life was being experienced, and participants with technology, the internet, and social media became indispensable. This cross-sectional survey was conducted in all 8 divisions of the country in which, 2926 out of 3,500 yielding a response rate of about 84%.

### Measures

A self-administered online survey was conducted using social media, in which 2926 out of 3,500 respondents correctly completed questionnaires. Anxiety and fear were determined by answering 'yes' to the question "*What kind of psychological problem do you feel for extensive use of the Internet*" while Physical issues were those answered relating to neck pain, headaches, and numbness to the question "*What kind of physical problem have you experienced from the extensive use of the Internet since COVID-19 lockdown*?". Social media addiction was those that to the questions "*Do you feel to urge use social media more and more*?", "*Do you become restless or feel troubled if unable to use social media*?", "*Do you spend a lot of time thinking about social media or planning to use social media*?" and "*Do you use social media so much that it could cause a negative impact on your job or studies*?" and social media hours was determined by the responded to the question,"How many hours do you spend daily on using social media?".

The data were collected over six weeks, starting from week 1 (2nd-8th September); followed by week 2 (9th-15th September), week 3 (16th-22nd September), week 4 (23rd-29th September), week 5 (30th-6th October) and week 6 (7th -13th October) in that order. Variables considered in the data collection included: gender, age, education level, work time and leisure time. A structured questionnaire was used for that purpose. Each variable in the questionnaire was coded; for instance, for gender, the male was coded "1", the female was coded "2", and the other was coded "88". For the level of education, Under SSC, SSC or equivalent, HSC or equivalent, Graduate, Postgraduate, Doctorate, Post Doctorate and Other were coded "1", "2", "3", "4", "5", "6" and "7, respectively.

### Ethics

The Institutional Review Board of the International University of Business, Agriculture and Technology (IUBAT), Dhaka, Bangladesh, granted permission for this study (IUBAT/AR/2021/002). The study followed the principles of the Helsinki Declaration, as updated in Fortaleza. Before completing the questionnaire, all participants were informed about the study's

specific goal. Data collected from the field was treated with high confidentiality, and prior to data collection, the participants were informed of the confidentiality of the information they provide. Verbal and written consent was obtained from all the participants, and the participants' confidentiality and privacy were maintained.

Respondents were permitted to complete the survey only once to avert repeated responses and to ensure that the data were valid to some extent. This was because data were limited to their IP addresses and device. Respondents had the option of terminating the survey at any time they desired. Furthermore, we ensured that the data were anonymous and confidential.

## Statistical analysis

Smart PLS 3 was used to analyse this study through a Partial Least Square Structural Equation Modelling (PLS-SEM) approach. There are several reasons for choosing this approach. These include its ability to deal with complex multivariate models and variables with different scales, explore theories, and test multiple mediators simultaneously [30]. We used the SmartPLS to have the individual parameters and the significance level [39]. In this study, the first step was to create groups according to the categorical variables of interest, including age, gender, education, working time and leisure time and this was followed by data analysis of the measurement reliability, validity for the latent reflective construct and social media addiction were examined by several indicators as suggested by Malak and colleagues [40]. Next, the second stage involved assessing the structural model correlations and hypotheses testing with significance levels. Model estimation was conducted with r2, Q2 and effect size describes the path effect from exogenous construct to endogenous construct [41]. Next, the path coefficients of the groups were analysed to determine if they were significantly diverse from each other based on the guidelines proposed by Henseler et al. [42]. To enable figures using latent constructs, a mean score of the four measurement items of social media addiction was created. Graphical representations using SPSS statistical software was used in order to further understand the interaction effects and the changes of the endogenous variables over time, and this was carried out by creating a mean score of the four measurement items of social media addiction and the variable were classified into five values, which are: 0, 0.25, 0.5, 0.75. 1.

## Results

Two thousand nine hundred and twenty-six Bangladeshi respondents participated in this study, and the breakdown of their characteristics are shown in Table 1 where 60.7% respondents were male and 39.3% were female. Table 2 presents the hypothesis testing, path

**Table 1. Characteristics of study participants (n = 2926).**

| Variable | Mean ± SD (min-max), or n (%) |
|---|---|
| Age | 33.2±6.7 (18, 65) |
| Education | 4.2±0.8 (1, 7) |
| Work Time | 2.7±1.2 (1, 8) |
| Leisure Time | 3.8±1.8 (1, 9) |
| Time in weeks | 3.3±1.7 (1, 6) |
| **Gender (n = 2920)** | |
| Male | 1776 (60.7) |
| Female | 1150 (39.3) |

N = 2926, otherwise in parentheses; min = minimum; max = maximum.

**Table 2. Hypothesis testing, path coefficients and their 95% Confidence Intervals (CIs).**

| Hypothesis | Relationship | Path coefficient (β) | t-value | Lower limit | upper limit | P-value |
|---|---|---|---|---|---|---|
| H1 | Anxiety -> Social Media Addiction | 0.19 | 6.83 | 0.15 | 0.24 | <0.001 |
| H2 | Fear -> Social Media Addiction | - 0.12 | 7.00 | -0.14 | -0.09 | <0.001 |
| H3 | Social Media Addiction -> Social Media Hours | 0.22 | 7.03 | 0.16 | 0.27 | <0.001 |
| H4 | Social Media Addiction -> Physical issues | 0.44 | 20.37 | 0.41 | 0.48 | <0.001 |
| H5 | Social Media Hours -> Physical issues | 0.07 | 3.23 | 0.03 | 0.10 | 0.001 |
| Control Variables | Age -> Physical issues | 0.01 | 0.83 | -0.01 | 0.04 | 0.203 |
| | Age -> Social Media Addiction | 0.03 | 1.61 | -0.01 | 0.06 | 0.053 |
| | Age -> Social Media Hours | -0.04 | 1.85 | -0.08 | -0.01 | 0.032 |
| | Edu -> Physical issues | -0.05 | 3.24 | -0.08 | -0.03 | 0.001 |
| | Edu -> Social Media Addiction | -0.06 | 3.07 | -0.09 | -0.03 | 0.001 |
| | Edu -> Social Media Hours | 0.14 | 6.06 | 0.10 | 0.17 | <0.001 |
| | Gender -> Physical issues | -0.03 | 2.63 | -0.05 | -0.01 | 0.004 |
| | Gender -> Social Media Addiction | -0.03 | 2.26 | -0.05 | -0.01 | 0.012 |
| | Gender -> Social Media Hours | -0.02 | 1.49 | -0.04 | 0.00 | 0.068 |
| | Leisure Time -> Physical issues | -0.21 | 7.91 | -0.26 | -0.17 | <0.001 |
| | Leisure Time -> Social Media Addiction | -0.08 | 2.56 | -0.13 | -0.03 | 0.005 |
| | Leisure Time -> Social Media Hours | -0.38 | 13.90 | -0.42 | -0.33 | <0.001 |
| | Work time -> Physical issues | 0.56 | 15.75 | 0.51 | 0.62 | <0.001 |
| | Work time -> Social Media Addiction | 0.66 | 15.48 | 0.59 | 0.73 | <0.001 |
| | Work time -> Social Media Hours | 0.76 | 20.10 | 0.70 | 0.82 | <0.001 |
| **Model Statistics** | | | | **$R^2$** | **$Q^2$** | |
| | Physical issues | | | 0.75 | 0.57 | |
| | Social Media addiction | | | 0.60 | 0.36 | |
| | Social Media hours | | | 0.57 | 0.57 | |

$R^2$ (R-Squared/Coefficient of determination) and $Q^2$ (The predictive relevance),

coefficients and their corresponding 95% Confidence Intervals (CIs), and the result suggests that the proposed model is well suited for confirming and explaining the positive effect of anxiety on social media addiction, as suggested in H1. Fear was negatively associated with social media addiction (β = -0.12, t = 7.00, p<0.001). H3 showed that social media addiction has a positive and significant effect on social media hours (β = 0.22, t = 7.03, p<0.001), similarly, a H4 showed a significant positive effect of social media addiction on physical issues (β = 0.44, t = 20.37, p<0.001). H5 revealed social media hours were significantly and positively related to physical issues (β = 0.07, t = 3.23, p<0.001), and all hypothesis reported in Table 2 was supported except H2.

The path coefficient for the five hypotheses differs statistically except for the effect of gender on social media hours and age on physical issues and social media addiction (see Table 2 for details). In this study, the value of $R^2$ on physical issues was substantially (75%), social media addiction was substantially (60%), and social media hours were higher than moderate (57%) because, in PLS-SEM, the $R^2$ value of 0.60 would be considered as substantial, 0.33 could be classified as mode. In contrast, 0.19 could be considered as weak [43]. The values of Q2 presented in Table 2 were higher than zero, which implies that this model has predictive relevance [43].

Table 3 presents hypothesis testing and path coefficient for time interactions. To test the moderating role of time after the Covid lockdown, the variable time (in weeks) was added to the model. Again, the model had good predictive accuracy ($R^2$>50%), and predictive relevance

**Table 3. Hypothesis testing, path coefficients and their 95% Confidence Intervals (CIs) for time interaction.**

| Hypothesis | Relationship | Path coefficient (β) | t-value | Lower limit | upper limit | P-value |
|---|---|---|---|---|---|---|
| H6a | Time * Anxiety--> Social Media Addiction | 0.06 | 3.84 | 0.03 | 0.08 | <0.001 |
| H6b | Time * Fear--> Social Media Addiction | 0.01 | 0.35 | -0.02 | 0.03 | 0.362 |
| H6c | Time * Social Media Addiction-->Social Media Hours | 0.04 | 2.50 | 0.01 | 0.07 | 0.007 |
| H6d | Time * Social Media Addiction--> Physical Issues | -0.02 | 2.09 | -0.04 | -0.01 | 0.020 |
| | **Model Statistics** | | | $R^2$ | $Q^2$ | |
| | Physical issues | | | 0.75 | 0.57 | |
| | Social Media addiction | | | 0.61 | 0.36 | |
| | Social Media hours | | | 0.58 | 0.57 | |

$R^2$ (R-Squared/Coefficient of determination) and $Q^2$ (The predictive relevance),

($Q^2 > 0$) (see Table 3 for details), and hypothesis 6a revealed that the interaction term of time and anxiety has a positive effect on social media addiction (β = 0.06, p<0.001). Hypothesis 6c showed that the interaction term of time and social media addiction was significantly related positively related to social media hours (β = 0.04, p<0.01), and the result in hypothesis 6d suggests that the proposed model is well suited for confirming and explaining the positive effect of the interaction term of time and social media addiction on physical issues and all hypothesis reported in Table 3 was supported except hypothesis 6b. We found a positive association between anxiety, social media addiction and social media hours and the physical issue was stronger over time (see Fig 2a, 2c and 2d: Moderation Effects), and fear and social media addition so were not statistically significant, and time did not influence the relationship (see Fig 2b: Moderation Effects).

As shown in the Fig 3a (Time* Anxiety → Social Media Addiction), participants who reported anxiety always had higher levels of social media addiction. Considering the fluctuation across the six weeks, the social media addiction of participants who had anxiety increased over time, and the social media addiction of participants who did not have anxiety decreased over time (mean = 0.77, SD = 0.32). The social media addiction level tends to be the same between the beginning and the end of the six weeks (see Fig 3b: Time* Fear →Social Media Addiction). Social media addiction was split into low (< = 0.75), which accounted for 38%, and high (= 1), which accounted for 62%. High social media addiction always had higher social media hours (Fig 3c: Time* Social Media Addiction → Social Media Hours) and physical issues (Fig 3d: Time* Social Media Addiction → Physical Issues) across the weeks, which confirms the result of H4.

Table 4 presents the Heterotrait-Monotrait Ratio (HTMT) results, and Table 3 indicates no discriminant validity problems according to the $HTMT_{0.85}$ criterion, indicating that there are no overlapping items from the respondents and the instrument used in this study has no problem in establishing the discriminant validity. The Cronbach's Alpha and Composite Reliability were 0.78 and 0.86, respectively. These were both greater than the cut-off points of 0.6, confirming the reliability of the measurements. To achieve convergent validity, factor loadings should be greater than 0.7 [44]. The factor loadings of the four items relating to social media addiction were, "*Do you spend a lot of time thinking about social media or planning to use social media?*" "*Do you feel urges to use social media more and more?*" "*Do you become restless or feel troubled if unable to use social media?*" and "*Do you use social media so much that it could cause a negative impact on your job or studies?*" were 0.83, 0.81, 0.73 and 0.72, respectively. The Average Variance Extracted (AVE) of the latent construct of social media addiction was 0.60, which was greater than the cut-off point of 0.50. Therefore, convergent validity was achieved. The

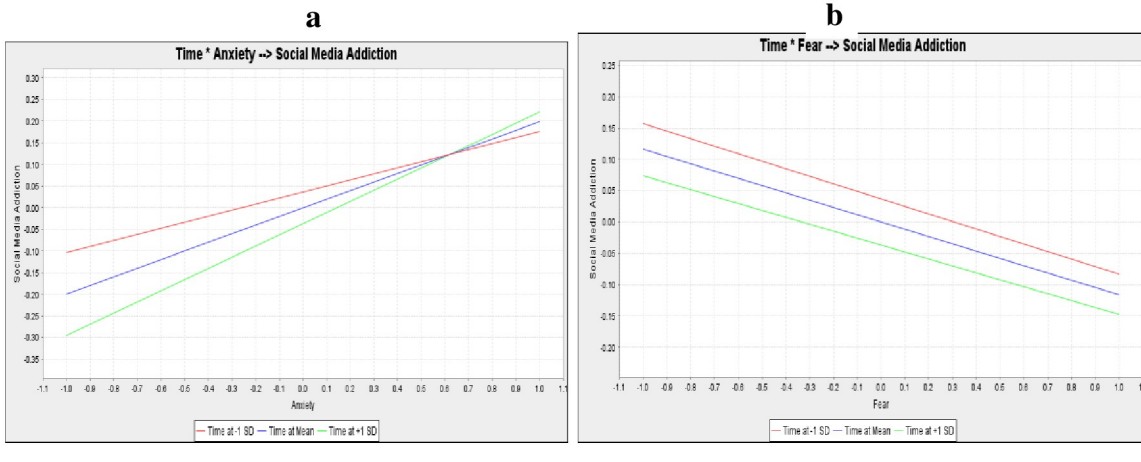

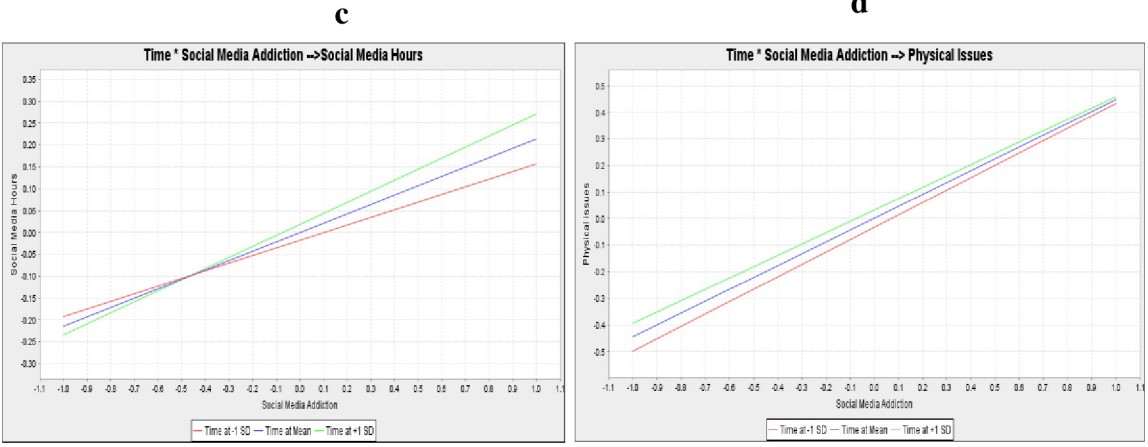

**Fig 2. (a-d): Moderation effects.**

heterotrait-monotrait ratio of correlations (HTMT) values ranged from 0.01 to 0.84 across all single-indicator constructs in discriminant validity (see Table 2 for details). The reflective latent construct was lower than the cut-off point of 0.9 [45]. To examine the common method bias, Variance Inflation Factor (VIF) should be lower than 3.3 [46], and all VIFs ranged from 1.0 to 1.36.

To assess the formative construct, weights of the indicators should be significant without collinearity problems [47]. The indicators were all significant (p<0.05) without collinearity issues (VIF<3.3): back pain (weight = 0.40, t = 15.65, p<0.001, VIF = 2.36), numbness (weight = 0.45, t = 18.20, p<0.001, VIF = 1.90), and headaches (weight = 0.28, t = 10.69, p<0.001, VIF = 2.18).

## Discussion

As addiction is defined as an irrepressible urge that is often accompanied by loss of control, internet addiction leads people to create problems from their uncontrollable abuse of Internet usage, which is related to other pathologies like depression, loneliness and social anxiety [48]. This current study aimed to examine how social media addiction and negative emotions influence health issues after the COVID-19 pandemic lockdown among Bangladeshi workers.

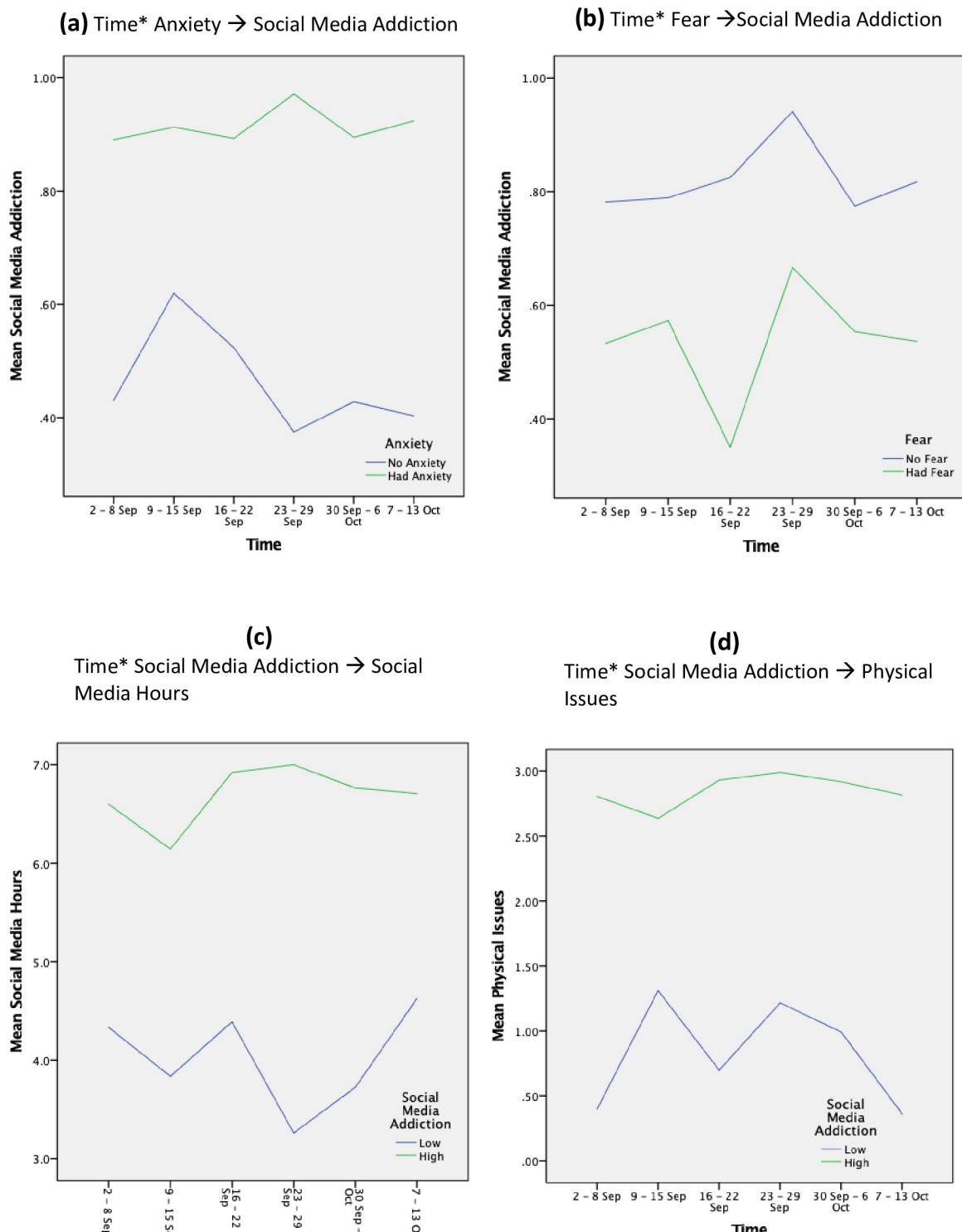

**Fig 3. Interaction effects and the changes of the endogenous variables over time.**

**Table 4. Heterotrait-Monotrait Ratio (HTMT).**

| | 1 | 2 | 3 | 4 | 5 | 6 | 7 | 8 | 9 |
|---|---|---|---|---|---|---|---|---|---|
| 1. Age | - | | | | | | | | |
| 2. Anxiety | 0.02 | - | | | | | | | |
| 3. Edu | 0.40 | 0.24 | - | | | | | | |
| 4. Fear | 0.01 | 0.16 | 0.01 | - | | | | | |
| 5. Gender | 0.22 | 0.19 | 0.20 | 0.08 | - | | | | |
| 6. Leisure Time | 0.01 | 0.31 | 0.21 | 0.02 | 0.15 | - | | | |
| 7. Social Media Addiction | 0.06 | 0.68 | 0.20 | 0.34 | 0.25 | 0.53 | - | | |
| 8. Social Media Hours | 0.06 | 0.67 | 0.29 | 0.24 | 0.21 | 0.32 | 0.72 | - | |
| 9. Work time | 0.04 | 0.63 | 0.28 | 0.22 | 0.24 | 0.74 | 0.84 | 0.68 | - |

Our study found that the anxiety level of the participants increased over the period of six weeks with the increase in social media addiction, which is in consonance with a past study from Turkey [49], which revealed that University students' social anxiety and happiness significantly forecast their addiction to social media. It is also consistent with several other previous studies [50–52]. This finding may be attributed to the fact that people who have communication difficulties in social environments and opt to create this kind of social interaction by the use of internet tools portray characteristics of social anxiety [53]. This is further buttressed by a past study [49] which found that happiness significantly forecasts university students' problematic internet use. They espoused that people who are content in their social environment and worry less about being evaluated in this environment normally do not seek different online communication tools—consequently, the possibility of their being addicted to social media declines.

A study in Bangladesh [54] aimed to assess the prevalence of anxiety among Bangladeshi individuals during the COVID-19 pandemic, vis-a-vis social media exposure (SME) and electronic media exposure (EME), backed the findings of the present study. Another past study revealed that perceived feelings of loneliness predicted both excessive social media use and anxiety, with excessive social media use also increasing anxiety levels [21]. In recent times, the use of social media has been highly lauded to receive health and safety information and ensure that social contacts are maintained to deal with the isolation of the pandemic [55]. Possibly due to the distressing situation, experts have suggested social media used to be a transient means of recovery from distress and as a coping strategy. This needs to be conscientiously managed to deal with loneliness and negative emotions [56]. Consequently, social media and virtual communities enable users to interact with other individuals, strengthen relationships, publicise content, portion out common interests, experiences, and emotions (e.g., [57]), and also enhance their engagement in digital platforms [58]. Nonetheless, there is the risk of social media involvement becoming excessive or dysfunctional by activating a behaviour–reward feedback loop [59] which strengthens negative moods and supports a vicious use of social media.

Surprisingly, the present study found a negative association between fear and social media addiction. The study showed that social media addiction decreases users' fear level, and fear level decreases with longer social media hours over the period of six weeks. This might be considered as the benefits of information exchange and peer support. Social media users can receive psychological support and advice from the people who are connected with them, which may help reduce their fear level. Thus, social interaction through social media, might positively affect the belief, ideas, and thoughts regarding the COVID-19 pandemic, social media use, which can aid build bridging, bonding, and maintained social capital [60].

As the strongest factor, "working hours" is positively correlated with social media hours, social media addiction, and physical problems. The significance of this finding is that working professionals who are involved in social media usage during working hours are likely to be pre-occupied with sustaining a sustained link with social media to make sure that they take part in all rewarding experiences being shared on these platforms. Eventually, such people exhibit decreased work efficiency (both decisional and action) and a reduction in work performance. Accordingly, our finding could be explained through the tenets of the limited capacity model [61], because of the fact that internet usage may burden the capacity of an individual to process information. Because of this burdened capacity, such users would not efficiently process work-related information and delay making decisions or executing work-related tasks. That is to say, working professionals' internet usage during working hours may hinder their intellectual processing ability and re-direct them from achieving their primary work tasks, resulting in reduced reported work performance decrement. Our finding aligns with that of past studies, which indicate that daily social media use during working hours is a distraction which has a negative impact on employees' work performance [62]. The significant association between social media usage and hesitation is a new addition to the extant literature. It reveals that social media usage during working house can impact the decision-making ability and work efficiency of an individual, which, to our knowledge, has not been examined before.

On the contrary, other control variables such as leisure time, gender, education level and age are directly or indirectly negatively associated with physical issues. The study found that social media users with longer leisure time, higher education, and female have fewer physical problems caused by social media usage. Our finding of the association of social media users with longer leisure time and physical activity demonstrates that the there is a greater likelihood of engaging against not engaging in all three types of physical activity among internet users than among those who do not use the internet. Moreover, the results show that, contradictory to presumptions that spending a long time sitting at the computer may result in a sedentary lifestyle; weekly hours spent online were not significant predictors of the likelihood of engaging versus not engaging in physical activity. The results suggest a possible relationship between other Internet use components and physical activity.

Our finding of the association between social media users with higher education and physical activity suggests that there is a correlation between internet usage for studying and physical activity on the one hand, and between internet usage and social media on the other, suggesting that both studying and social media empower individuals to access information which could help them to develop particular plans for physical activity customized to their needs, perceptions, and abilities, whether strenuous or moderate. With regards to physical exercise to strengthen muscles, the particular digital uses, seeking information and playing games were identified as those which correlated with this type of physical activity. Looking for information is another activity which suggests a need for acquisition of knowledge and being competent in making plans about physical activity [63] independently.

The association between female social media users and physical activity showed that females had a stronger insight of ease of use, compatibility, relative advantage, and risk when they use social media, in comparison to men. This findings from the study also have been supported by some previous studies as well [64, 65]. More recent research, for instance, one by Lin and Wang [66] aimed to explain the differences in gender in information-sharing behavior on social networking sites. To accomplish this, a comparative theoretical model of information sharing between genders was established. Consistent with past research, analysis revealed a greater importance about that privacy risk, social ties, and commitment for women than men, because attitude towards information sharing impacts people's intention to share information more strongly for women than it does for men. Another recent research [67] made an attempt

to examine gender differences in the use of social media by investigating adolescents aged 13–18 years in the U.S. and UK. Results showed that adolescent girls spent more time on smartphones, social media, texting, and general computer use, compared with boys. However, no further assessment was conducted to ascertain how much of this time was spent to plan an activity.

Moreover, older people are less addicted to social media, causing fewer physical disorders. We observed that females were less addicted to social media than their male counterparts, and therefore are particularly less vulnerable regarding physical issues caused by social media addiction. Hence, it can be summarized that generally, males with long working hours spend more time on social media and become addicted to it and consequently become victims of several physical disorders.

As discussed above, there exists is a positive association between social media usage hours and social media addiction level. Therefore, we can combine the terms like social media addiction to discuss the effects of social media usage hours and its effect on physical issues and the effect of social media addiction on physical issues. Our study found that several physical problems related to physical issues increase with the increase of social media addiction levels. Several studies support the findings of the present study. For instance, a study in Iran [68] proposed a possible psychopathology mechanism to elucidate psychological distress among Iranian young adults during the COVID-19 public health crisis. There was a significant association between problematic social media use and psychological distress, both directly and indirectly.

Moreover, a study [69] conducted in a Tech based company in India also revealed that employees engaged in too much social media usage were having sleep deprivation; eye strain; feeling of resentment; lack of depth in the relationships; compromise with the work quality, and a distraction from work. Another study among social media users observed a number of physical, psychological and behavioural issues. Frequently seen physical problems included a strain on eyes, neck pain, back pain, headache, watering of eyes, wrist and shoulder pain, which were consistent with other studies [70]. However, some exogenous variables were also considered to see if any external factors influence the association between Internet addiction and physical issues in this study. The study found that "working hours" only positively correlated with physical issues.

There is a stronger role that social media could play, that could enable us to treat socially-shaped diseases like obesity, depression, diabetes, heart disease, and other mental illnesses. A past study [71] outlined how social network thinking is growing, and described several current uses of social media in healthcare before describing how we could harness the understanding of social networks and media for this stronger role of treating socially physical and psychological diseases though on the platform, obsessive users have a higher prevalence of social encounters [72]. However, every aspect of daily living has been disrupted by the COVID-19 pandemic, giving rise to forced isolation and practising social distancing, economic difficulty, and fears of being infected by a potentially fatal illness, that could make a person to feel helpless and hopeless [73]. Past research [74] expressed concern about the way and manner individuals have endured in the past so as to identify strategies which could be especially successful in controlling health issues and developing resilience during critical times.

Notwithstanding the fact that this current study was systematically designed, it was limited in a number of ways. Firstly, our results were based on only workers in government and non-government companies. Secondly, the study covered only workers aged between 25 and 45 years. Due to the fact that the PLS-SEM technique is new and easy to implement, we can extend the study to assess the influence of the pandemic lockdown on other aspects of life other than health, people aged outside the age bracket 25–45 years. Thirdly, despite the fact

that research on the PLS-SEM method has gained popularity during in the last ten years, there are sufficient research opportunities on subjects like mediation or multi-group analysis, which necessitate further investigation.

## Conclusion

The study is potentially significant because it will offer social media users, healthcare workers, and policymaker's insights into the adverse effect of addictive social media use. Most importantly, it will highlight the association of social media addiction with different issues related to our work, health, education, age, working hours, leisure time and gender, the very important issues at the centre of life.

## Supporting information

**S1 File.**
(ZIP)

## Author Contributions

**Conceptualization:** Dewan Muhammad Nur –A Yazdani, Tanvir Abir.

**Data curation:** Tanvir Abir, Kingsley Emwinyore Agho.

**Formal analysis:** Tanvir Abir, Kingsley Emwinyore Agho, Shasha Wang.

**Methodology:** Tanvir Abir, Abdullah Al Mamun, Kingsley Emwinyore Agho, Shasha Wang.

**Supervision:** Noor Raihani Zainol, Kaniz Kakon.

**Validation:** Kingsley Emwinyore Agho.

**Writing – original draft:** Dewan Muhammad Nur –A Yazdani, Tanvir Abir, Yang Qing, Jamee Ahmad.

**Writing – review & editing:** Dewan Muhammad Nur –A Yazdani, Tanvir Abir, Yang Qing, Jamee Ahmad, Abdullah Al Mamun, Noor Raihani Zainol, Kaniz Kakon, Shasha Wang.

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
