## [Decision Letter · Decision Letter 0]

13 Apr 2022

PONE-D-22-03615Social media addiction and emotions during the disaster recovery period – The moderating role of post-COVID timing.PLOS ONE

Dear Dr. Nur -A Yazdani,

Thank you for submitting your manuscript to PLOS ONE. After careful consideration, we feel that it has merit but does not fully meet PLOS ONE’s publication criteria as it currently stands. Therefore, we invite you to submit a revised version of the manuscript that addresses the points raised during the review process. The referees require a MAJOR REVISION of your paper. If you are prepared to undertake the work required, I would be pleased to reconsider my decision. However, I would like to ask you to very carefully address the issues raised by the referees and revise your paper accordingly.

We look forward to receiving your revised manuscript.

Kind regards,

Barbara Guidi

Academic Editor

PLOS ONE

Journal Requirements:

2.Please provide additional details regarding participant consent. In the ethics statement in the Methods and online submission information, please ensure that you have specified what type you obtained (for instance, written or verbal, and if verbal, how it was documented and witnessed). If your study included minors, state whether you obtained consent from parents or guardians. If the need for consent was waived by the ethics committee, please include this information.

Reviewers' comments:

Reviewer's Responses to Questions

**Comments to the Author**

1. Is the manuscript technically sound, and do the data support the conclusions?

Reviewer #1: Yes

Reviewer #2: Partly

2. Has the statistical analysis been performed appropriately and rigorously? 

Reviewer #1: Yes

Reviewer #2: No

3. Have the authors made all data underlying the findings in their manuscript fully available?

Reviewer #1: Yes

Reviewer #2: No

4. Is the manuscript presented in an intelligible fashion and written in standard English?

Reviewer #1: Yes

Reviewer #2: No

5. Review Comments to the Author

Reviewer #1: In this paper, the authors describe the impact of social media addiction during the COVID pandemics. The paper is well written, and contains relevant finding. I have some suggestions for the authors:

- Avoid using wording like "present" when you talk about a specific point in time (in few years, it is no more "present"), but instead use the actual year (2022).

- The formulation of Hypothesis 6 is unclear. To begin with, you probably meant "over time", because "overtime" is a noun and it does not fit well in each sentence. Additionally, I don't get what the sentences like "the relationship between anxiety and social media addiction will be more substantial overtime;" mean. Do you mean that the relationships you identified in the previous hypotheses will get stronger over time? Or maybe do you mean that "if you become more anxious you will become more addicted to social media"? I think you should state the hypothesis in a much clearer way.

- I think the hypotheses should be part of the framework, to give it the correct importance in the paper.

- Consider citing the following paper:

Nasti, Lucia, Andrea Michienzi, and Barbara Guidi. "Discovering the Impact of Notifications on Social Network Addiction." International Symposium: From Data to Models and Back. Springer, Cham, 2020.

I think you will find the findings shown in this paper relevant to your work.

Reviewer #2: Thank you for the opportunity to read this manuscript about problematic social media use during the COVID pandemic. I appreciate the efforts made by the authors to recruit this sample, as the sample size is remarkable. However, there is much work to do to improve this paper.

- I encourage the authors to move away from the addiction framework in presenting findings related to problematic social media use PSMU. Authors recommend a dose of skepticism towards the idea that frequent social media use might indicate a disorder or even only a mere symptom of a different primary condition (see, for a summary, Casale & Banchi, 2017). I also encourage the authors to cite the conceptualization of PSMU that you use

- The authors argue that social media addiction leads to psychological and physical consequences (e.g., lines 86-95). However the vast majority of the cited studies used cross-sectional designs which do not allow to draw causal inferences.

_ Please, check the following sentence: "social media usage hours have been found to be strongly associated with creating social media addiction. Also consider that the correlation between time spent on social media and problematic social media use has been often found to be low.

- H1 and H2 deal with the link between anxiety/fear and PSMU. A lot of previous studies focused on this link, but the authors did not present them in their introduction.

- H3: social media addiction increases with social media usage hours. However, Figure 1 shows the opposite direction

- In order to test H4 and H5 the authors should have adopted a longitudinal design. I encourage the authors to reformulate their hypotheses by recplacing "increase" with "is correlated with". Similarly, the authors should not use terms as "independent" and "dependent" variables as they did not use an experimental design.

- There is no need to provide a definition of "conceptual framework".

- The Method section paragraphs should be presented in the following order: 1. Participants. 2. Measures 2. Procedure. 4. Ethics. 5. Statistical analyses.

- The Measures (including the psychometric adaptation of their versions in Bangladesh) should be better described. Provide the name for IDS-15 and SMD. Why the authors also used the GPIUS? I was not able to find information about the measures used to assess anxiety and fear.

- Table 2: " à" ?

- The authors should then provide: descriptive statistics; bivariate correlations among the study variables; results from the SEM. There is no need to present results separately by each week. Moreover, the SEM results are not properly reported.

- It is impossible to evaluate the discussion section until the data are properly analyzed. In any case, the Discussion section is full of not-relevant sentences (e.g., lines 454-463).

By the way, the authors found that women are less addicted to social media than men. This result is in contrast with previous findings (see for a meta-analysis Su et al., 2020) and interpretation should be provide.

6. PLOS authors have the option to publish the peer review history of their article (what does this mean?). If published, this will include your full peer review and any attached files.

Reviewer #1: No

Reviewer #2: No

---

## [Author Response · Author response to Decision Letter 0]

29 Jul 2022

Editorial Team

July 2022

Manuscript ID: PONE-D-22-03615

“Social media addiction and emotions during the disaster recovery period – The moderating role of post-COVID timing”

Dear Editor-in-chief, 

Thank you for allowing us to revise and re-submit our manuscript. We are incredibly grateful for your insightful feedback that has strengthened our manuscript. 

We have addressed all the comments raised by the 1st reviewer in the revised version of the manuscript. All changes made to the revised manuscript have been in track changes. Below, we provide a point-by-point response to the comments and linked our responses to the appropriate sections in the revised manuscript. We hope that the revisions made have properly addressed the concerns, and that our revised manuscript is acceptable for publication in the journal.

Reviewer 1

In this paper, the authors describe the impact of social media addiction during the COVID pandemics. The paper is well written, and contains relevant finding. I have some suggestions for the authors:

 Comment Feedback

1 Avoid using wording like "present" when you talk about a specific point in time (in few years, it is no more "present"), but instead use the actual year (2022). 

---Done. The Section was revised.

2 The formulation of Hypothesis 6 is unclear. To begin with, you probably meant "over time", because "overtime" is a noun and it does not fit well in each sentence. Additionally, I don't get what the sentences like "the relationship between anxiety and social media addiction will be more substantial overtime;" mean. Do you mean that the relationships you identified in the previous hypotheses will get stronger over time? Or maybe do you mean that "if you become more anxious you will become more addicted to social media"? I think you should state the hypothesis in a much clearer way. 

---The sentences were modified with the word ‘over time’ in space of ‘overtime’.

3 I think the hypotheses should be part of the framework, to give it the correct importance in the paper. 

---The hypotheses were replaced under the framework.

4 Consider citing the following paper:

Nasti, Lucia, Andrea Michienzi, and Barbara Guidi. "Discovering the Impact of Notifications on Social Network Addiction." International Symposium: From Data to Models and Back. Springer, Cham, 2020.

I think you will find the findings shown in this paper relevant to your work. 

---The information from the paper was added to the manuscript and the reference list has been updated.

Reviewer 2:

Thank you for the opportunity to read this manuscript about problematic social media use during the COVID pandemic. I appreciate the efforts made by the authors to recruit this sample, as the sample size is remarkable. However, there is much work to do to improve this paper.

 Comment Feedback

1 I encourage the authors to move away from the addiction framework in presenting findings related to problematic social media use PSMU. Authors recommend a dose of skepticism towards the idea that frequent social media use might indicate a disorder or even only a mere symptom of a different primary condition (see, for a summary, Casale & Banchi, 2017). I also encourage the authors to cite the conceptualization of PSMU that you use. 

---References has been shown for the conceptualization of the PSMU used in the study.

2 The authors argue that social media addiction leads to psychological and physical consequences (e.g., lines 86-95). However the vast majority of the cited studies used cross-sectional designs which do not allow to draw causal inferences. 

---This argument was supported by many articles cited within the manuscript especially from 55 – 67 in the references.

3 Please, check the following sentence: "social media usage hours have been found to be strongly associated with creating social media addiction. Also consider that the correlation between time spent on social media and problematic social media use has been often found to be low. 

---Done. The sentence was restructured and correlation was shown.

4 H1 and H2 deal with the link between anxiety/fear and PSMU. A lot of previous studies focused on this link, but the authors did not present them in their introduction. 

---More references has been added to support hypotheses H1 and H2.

5 H3: social media addiction increases with social media usage hours. However, Figure 1 shows the opposite direction 

---H3: Social media addiction are associated with social media usage hours.

6 In order to test H4 and H5 the authors should have adopted a longitudinal design. I encourage the authors to reformulate their hypotheses by recplacing "increase" with "is correlated with". Similarly, the authors should not use terms as "independent" and "dependent" variables as they did not use an experimental design. 

---The word ‘increases’ was replaced with ‘is correlated with’. Also the terms "independent" and "dependent" variables were changed as per the discussion.

7 There is no need to provide a definition of "conceptual framework". 

---Done. The Section was revised.

8 The Method section paragraphs should be presented in the following order: 1. Participants. 2. Measures 2. Procedure. 4. Ethics. 5. Statistical analyses. 

---The method section was restructured.

9 The Measures (including the psychometric adaptation of their versions in Bangladesh) should be better described. Provide the name for IDS-15 and SMD. Why the authors also used the GPIUS? I was not able to find information about the measures used to assess anxiety and fear. 

---We have provided some information about the IDS-15 and SMD and why the GPIUS was used. Furthermore the measures used to assess anxiety and fear was also indicated within the manuscript and marked in yellow.

10 Table 2: " à" ? 

---Corrected.

11 The authors should then provide: descriptive statistics; bivariate correlations among the study variables; results from the SEM. There is no need to present results separately by each week. Moreover, the SEM results are not properly reported. 

---Agreed and we have produced Heterotrait-Monotrait Ratio (HTMT)correlation matrix and SEM results are now properly written.

12 It is impossible to evaluate the discussion section until the data are properly analyzed. In any case, the Discussion section is full of not-relevant sentences (e.g., lines 454-463). 

---The mentioned paragraph in the discussion section is revised as per the theme of the study.

13 By the way, the authors found that women are less addicted to social media than men. This result is in contrast with previous findings (see for a meta-analysis Su et al., 2020) and interpretation should be provide. 

---This finding of the study has been justified this with proper references.

 Comments to the Author Feedback

1 Is the manuscript technically sound, and do the data support the conclusions?

The manuscript must describe a technically sound piece of scientific research with data that supports the conclusions. Experiments must have been conducted rigorously, with appropriate controls, replication, and sample sizes. The conclusions must be drawn appropriately based on the data presented. Reviewer #1: Yes

Reviewer #2: Partly

2 Has the statistical analysis been performed appropriately and rigorously? Reviewer #1: Yes

Reviewer #2: No

3 Have the authors made all data underlying the findings in their manuscript fully available?

The PLOS Data policy requires authors to make all data underlying the findings described in their manuscript fully available without restriction, with rare exception (please refer to the Data Availability Statement in the manuscript PDF file). The data should be provided as part of the manuscript or its supporting information, or deposited to a public repository. For example, in addition to summary statistics, the data points behind means, medians and variance measures should be available. If there are restrictions on publicly sharing data—e.g. participant privacy or use of data from a third party—those must be specified. Reviewer #1: Yes

Reviewer #2: No

4 Is the manuscript presented in an intelligible fashion and written in standard English?

PLOS ONE does not copyedit accepted manuscripts, so the language in submitted articles must be clear, correct, and unambiguous. Any typographical or grammatical errors should be corrected at revision, so please note any specific errors here. Reviewer #1: Yes

Reviewer #2: No

---

## [Decision Letter · Decision Letter 1]

7 Sep 2022

Social media addiction and emotions during the disaster recovery period – The moderating role of post-COVID timing.

PONE-D-22-03615R1

Dear Dr. Nur -A Yazdani,

We’re pleased to inform you that your manuscript has been judged scientifically suitable for publication and will be formally accepted for publication once it meets all outstanding technical requirements.

Kind regards,

Barbara Guidi

Academic Editor

PLOS ONE

Additional Editor Comments (optional):

Reviewers' comments:

Reviewer's Responses to Questions

**Comments to the Author**

1. If the authors have adequately addressed your comments raised in a previous round of review and you feel that this manuscript is now acceptable for publication, you may indicate that here to bypass the “Comments to the Author” section, enter your conflict of interest statement in the “Confidential to Editor” section, and submit your "Accept" recommendation.

Reviewer #1: All comments have been addressed

2. Is the manuscript technically sound, and do the data support the conclusions?

Reviewer #1: (No Response)

3. Has the statistical analysis been performed appropriately and rigorously? 

Reviewer #1: (No Response)

4. Have the authors made all data underlying the findings in their manuscript fully available?

Reviewer #1: (No Response)

5. Is the manuscript presented in an intelligible fashion and written in standard English?

Reviewer #1: (No Response)

6. Review Comments to the Author

Reviewer #1: All comments have been addressed, therefore I recommend to publish the paper when the editor thinks so

7. PLOS authors have the option to publish the peer review history of their article (what does this mean?). If published, this will include your full peer review and any attached files.

Reviewer #1: No

---

## [Editor Report · Acceptance letter]

12 Oct 2022

PONE-D-22-03615R1 

Social media addiction and emotions during the disaster recovery period – The moderating role of post-COVID timing. 

Dear Dr. Nur -A Yazdani:

I'm pleased to inform you that your manuscript has been deemed suitable for publication in PLOS ONE. Congratulations! Your manuscript is now with our production department. 

Kind regards, 

on behalf of

Dr. Barbara Guidi 

Academic Editor

PLOS ONE